# A Rationale for the Activity of Bone Target Therapy and Tyrosine Kinase Inhibitor Combination in Giant Cell Tumor of Bone and Desmoplastic Fibroma: Translational Evidences

**DOI:** 10.3390/biomedicines10020372

**Published:** 2022-02-03

**Authors:** Alessandro De Vita, Silvia Vanni, Giacomo Miserocchi, Valentina Fausti, Federica Pieri, Chiara Spadazzi, Claudia Cocchi, Chiara Liverani, Chiara Calabrese, Roberto Casadei, Federica Recine, Lorena Gurrieri, Alberto Bongiovanni, Toni Ibrahim, Laura Mercatali

**Affiliations:** 1Osteoncology and Rare Tumors Center, IRCCS Istituto Romagnolo Per Lo Studio Dei Tumori (IRST) “Dino Amadori”, 47014 Meldola, Italy; alessandro.devita@irst.emr.it (A.D.V.); giacomo.miserocchi@irst.emr.it (G.M.); valentina.fausti@irst.emr.it (V.F.); chiara.spadazzi@irst.emr.it (C.S.); claudia.cocchi@irst.emr.it (C.C.); chiara.liverani@irst.emr.it (C.L.); chiara.calabrese@irst.emr.it (C.C.); lorena.gurrieri@irst.emr.it (L.G.); alberto.bongiovanni@irst.emr.it (A.B.); laura.mercatali@irst.emr.it (L.M.); 2Pathology Unit, Morgagni-Pierantoni Hospital, 47121 Forli, Italy; federica.pieri@auslromagna.it; 3Orthopedic Unit, Morgagni-Pierantoni Hospital, 47121 Forli, Italy; roberto.casadei@auslromagna.it; 4Medical Oncology Unit, Azienda Ospedaliera San Giovanni Addolorata, 00184 Roma, Italy; federica.recine@irst.emr.it; 5Osteoncology, Bone and Soft Tissue Sarcomas and Innovative Therapies Unit, IRCCS Istituto Ortopedico Rizzoli, 40136 Bologna, Italy; toni.ibrahim@ior.it

**Keywords:** bone sarcoma, giant cell tumor of bone, desmoplastic fibroma, primary culture, 3D scaffold, gene expression profiling, denosumab, lenvatinib, zebrafish

## Abstract

Giant cell tumor of bone (GCTB) and desmoplastic fibroma (DF) are bone sarcomas with intermediate malignant behavior and unpredictable prognosis. These locally aggressive neoplasms exhibit a predilection for the long bone or mandible of young adults, causing a severe bone resorption. In particular, the tumor stromal cells of these lesions are responsible for the recruiting of multinucleated giant cells which ultimately lead to bone disruption. In this regard, the underlying pathological mechanism of osteoclastogenesis processes in GCTB and DF is still poorly understood. Although current therapeutic strategy involves surgery, radiotherapy and chemotherapy, the benefit of the latter is still debated. Thus, in order to shed light on these poorly investigated diseases, we focused on the molecular biology of GCTB and DF. The expression of bone-vicious-cycle- and neoangiogenesis-related genes was investigated. Moreover, combining patient-derived primary cultures with 2D and 3D culture platforms, we investigated the role of denosumab and levantinib in these diseases. The results showed the upregulation of *RANK-L, RANK, OPN, CXCR4*, *RUNX2* and *FLT1* and the downregulation of *OPG* and *CXCL12* genes, underlining their involvement and promising role in these neoplasms. Furthermore, in vitro analyses provided evidence for suggesting the combination of denosumab and lenvatinib as a promising therapeutic strategy in GCTB and DF compared to monoregimen chemotherapy. Furthermore, in vivo zebrafish analyses corroborated the obtained data. Finally, the clinical observation of retrospectively enrolled patients confirmed the usefulness of the reported results. In conclusion, here we report for the first time a molecular and pharmacological investigation of GCTB and DF combining the use of translational and clinical data. Taken together, these results represent a starting point for further analyses aimed at improving GCTB and DF management.

## 1. Introduction

First reported in 1953 by Jaffe [1], giant cell tumor of bone (GCTB). also known as osteoclastoma, is described in the latest WHO version as an osteolytic bone tumor characterized by the presence of osteoclast-like multinucleated giant cells [2]. Although this locally aggressive and rarely metastasizing tumor is generally considered a benign neoplasm due to its indolent behavior, it often causes severe bone resorption as a result of RANK signaling promotion of multinuclear osteoclast generation [3]. In this regard, multinucleated giant cells (MGCs), which are similar in morphology and function to osteoclasts, are considered the main cause of bone damage by GTCB. MGCs are recruited and induced from mononuclear precursors of osteoclasts by the tumor stromal cells, which represent the neoplastic component of the tumor and often harbor a highly specific mutation in the histone variant H3.3 which is encoded by *H3F3A* [4]. GCTB occurs in young adults in epiphyseal–metaphyseal regions of the long bones, such as distal femur (26%), proximal tibia (20%) and distal radius (11%), while in iliac bone, spine and hand bones, cases are rarely reported [5,6,7,8]. The incidence of this bone sarcoma is estimated in almost 5% of all primary bone tumors, with a slight female predilection [9]. GCTB has a characteristic radiographic appearance, with an eccentric, osteolytic lesion arising in the medullary portion of bone and extending to the subchondral bone at the articular surface. Long-bone GCT is usually well-marginated, without peripheral sclerosis, and frequently shows focal cortical destruction. Intralesional mineralization and periosteal reaction are usually absent. Differently from giant cell osteosarcoma, in which malignant spindle cells produce osteoid in a background of giant cells, the presence of malignant stromal cells is indicative of GCT of bone [10]. While Enneking staging [11] and Campanacci grading system, the latter based on radiographic imaging [12], may be used to determine definitive management, clinical history and physical examination can be helpful in the differential diagnosis. Macroscopically, the lesion usually appears as a large (from a few cm to >15 cm) red-brown and friable mass with hemorrhagic and cystic areas, however in some cases they present as white and fleshy tumors. Yellow areas are representative of sheets of lipid-filled macrophages. While indolent tumors remain confined within the medulla, advanced tumors destroy bone and invade soft tissue, where recurrent tumors are often surrounded by a shell of bone [13]. Histologically, GCTB composes a mixture of mononuclear stromal cells and reactive osteoclastic giant cells expressing Receptor Activator of Nuclear factor Kappa-B (RANK) [14], the latter usually distributed evenly throughout the tumor and showing up to 100 nuclei. The mononuclear stromal cells typically show oval to reniform nuclei, which can also be spindle shaped, ill-defined cell borders and abundant cytoplasm. The histologic appearance of GCTB can be modified by secondary alterations, including fibrohistiocytic and xanthomatous patterns, fibrosis and hemosiderosis, coagulation necrosis, cystic degeneration, abundant woven bone, and rarely, cartilage formation. GCTB shows a wide spectrum of chromosomal aberrations, especially in chromosome 11p which is frequently altered, with telomeric fusions being the most frequent cytogenetic finding [15]. Moreover, alterations in the *c-myc*, *N-myc*, and *c-fos* oncogenes, as well as alterations in p53 in the metastatic foci, have been reported [16].

Metachronous and multicentric involvement by primary GCTB is a rare event occurring in less than 1% of cases. Metastases occur in only around 2% of cases and most are in the lungs, followed by bone, brain, kidney, adrenal gland, gastrointestinal tract, and skin. Patients with metastatic disease often have a long, indolent course. Despite this, GCTB prognosis is unpredictable, showing no correlation to either histologic grading or vascular invasion, with 25% dying of disease. However, tumors harboring clonal cytogenetic aberrations are thought to be more aggressive [10]. Currently, the main treatment option for GCTB is represented by surgery, typically curettage or en-bloc resection, with the possible local administration of adjuvants such as liquid nitrogen, phenol or polymethylmethacrylate [17,18]. Management of GCTB is particularly challenging due to the high recurrence rate after surgery. Recurrences are highly frequent in so-called high-risk GCTB, tumors characterized by extension into surrounding soft tissue, pathologic fracture, absence of local postsurgery adjuvant therapy, frequent recurrence and localization in the spine or sacrum [14]. Neoadjuvant treatment of GCTB with denosumab—a human monoclonal IgG2 antibody that inhibits activation and differentiation of osteoclast-like giant cells and consequent osteolytic damage by binding RANK-ligand [19]—can effectively reduce tumors to facilitate surgery or avoid the need for resection, but there is concern about local recurrence postsurgery. Definitive treatment of unresectable GTCB improves symptoms and blocks tumor progression [20,21]. However, long-term treatment leads to adverse events such as osteonecrosis and fractures.

Another benign bone sarcoma is represented by desmoplastic fibroma (DF) of bone, first reported in 1958 by Jaffe [22]. This very rare fibroblastic neoplasm shows locally aggressive behavior without metastatic propensity and usually occurs in young adults. This primary bone tumor typically consists of well-differentiated myofibroblasts surrounded by abundant collagenous tissue, generally lacking pleomorphism and atypical mitotic figures typical of fibrosarcoma [23]. The incidence of DF is estimated to be about 0.1% of all bone tumors, with a slight prevalence for males. The most common site of disease is the mandible (22%) and less frequently other bones such as femur (15%), pelvic bones (13%), radius (12%), and tibia (9%) [24]. Concerning etiology, no conclusive data are available, however, previous pathologic fractures are sometimes observed. Radiographically, this entity presents as a well-defined, lobulated and radiolucent lesion. Larger lesions may destroy cortical bone and extend into surrounding soft tissue, events which are well-appreciated by MRI. DF has a low signal intensity in T1- and T2-weighted MRI images, and may show increased uptake with bone scintigraphy or FDG PET. Diagnosis is based on histopathological features, including low-to-moderate cellularity and abundant collagen fibers. DF cells exhibit poorly defined borders and absence of nuclear atypia. Differential diagnosis includes low-grade fibrosarcoma, fibrous dysplasia, low-grade osteosarcoma, simple bone cyst, aneurysmal bone cyst, non-ossifying fibroma, eosinophilic granuloma, chondromyxofibroma and metastases [25]. Local recurrence after surgery is common (about 50% of cases), but decreases when resection is performed instead of excision or curettage [23]. Clinical manifestation reported in the latest WHO report described longstanding history of pain and bone deformity. The cornerstone of treatment is represented by surgery with wide resection or en bloc resection and reconstructive surgery with long-term follow-up [26]. Radiotherapy is reported as a therapeutic option, but its role is still debated [27,28]. Chemotherapy including vincristine, doxorubicin and dacarbazine has been described with unclear efficacy [26].

Although these bone sarcomas are considered benign lesions, given their nonmetastatic potential, their clinical manifestation leads to important bone resorption, strongly affecting the patients’ quality of life. For the above reasons, there is a pressing need to deepen their poorly understood biology and to identify new therapeutic paradigms for their clinical management. This work would provide the readership with new translational evidence about the role of the tumor microenvironment as a potential therapeutic target and would open the door for further analyses.

## 2. Materials and Methods

### 2.1. Ethical Statement and Case Series

All human samples were anonymized. The study involved four adult patients affected by GCTB and one affected by DF. IRST-Area Vasta Romagna Ethics Committee approved the study protocol, approval no. 4751, 31 July 2015. Good Clinical Practice standard operating procedures and 1975 Helsinki declaration were applied in the study. Informed consent for participation in the research study was obtained from each patient. Enrolment of patients started in January 2019 and ended in August 2021.

### 2.2. Histological Analyses

Histopathological features of tumor tissues were analyzed through hematoxylin-eosin (H&E) staining. Briefly, resected tumor tissue was fixed in 10% formalin, dehydrated and paraffin embedded. Then, 5 µm-thick slices were obtained and stained using standard techniques, as previously reported [29].

### 2.3. Tissues and RNA Extraction

Total RNA was isolated from about 50 mg of fresh-frozen tumor tissue and related healthy tissue from the surgical margin of each excised tumor. The tissue was homogenized with a homogenizer (IKA T18 Basic ULTRA-TURRAX) in TRIzol reagent (Invitrogen, Waltham, MA, USA) following manufacturer’s instructions. RNA was extracted with RNeasy Mini Kit (Qiagen, Germantown, MD, USA) and on-column DNA digestion was performed. RNA was checked for concentration and purity on a NanoDrop 2000 spectrophotometer (Thermo Scientific, Waltham, MA, USA).

### 2.4. Reverse Transcription and Real-Time Quantitative PCR (RT-qPCR)

First, 400 ng of total RNA was used to obtain cDNA with iScript cDNA Synthesis Kit (BioRad, Hercules, CA, USA). A negative control for each sample was performed by omitting reverse transcriptase. The online tool Primer-Blast provided by NCBI was used to design primers, spanning an exon-exon junction or flanking intron sequence/s to avoid genomic DNA contamination. The primers and probes sequences were as previously reported [30,31,32]. Taqman Universal PCR Master Mix (Applied Biosystems, Waltham, MA, USA) or SYBR Select Master Mix (Applied Biosystems, Waltham, MA, USA) were used for gene expression assay, with 400 nM final concentration of the corresponding forward and reverse primer (Sigma, Burlington, MA, USA) or Taqman probe, and 10 ng/μL final concentration of cDNA samples. Amplification was performed using 7500 Real-Time PCR System (Applied Biosystems, Foster City, CA, USA) with the following cycling conditions: 3 min at 95 °C, 45 cycles at 95 °C for 10 sec and 60 °C for 1 min. Technical triplicates were performed for each primer pair and sample. Negative control for each primer pair was analyzed to exclude genomic RNA amplification. Nontemplate control (NTC) was included to rule out genomic DNA presence in the qPCR reaction mix. To ensure that dimers of primers were not influencing the obtained fluorescence signals, melting curves were checked for each primer pair.

### 2.5. RT-qPCR Data Analysis

*B2M* expression was used as a reference gene to normalize differential expression of the target gene. The 2 ^−∆∆CT^ method was used to calculate fold change (FC) with the following steps: ∆C_T_ were calculated subtracting the C_T_ of the reference gene from the C_T_ of the target gene, both for the “tumor” sample and “healthy” sample. Then, ∆∆C_T_ was calculated.

### 2.6. Isolation of Patient-Derived GCTB and DF Primary Cells

Isolation and establishment of GCTB patient-derived primary cells were performed according to previously reported protocol [33,34]. In brief, surgical resected tumor tissue was finely cut with surgical scalpels and enzymatically digested overnight. The subsequent day, the obtained cell suspension was filtered. Then, isolated tumor cells were seeded in standard monolayer cultures or in collagen-based scaffolds. For the 2D culture model, a cell density of 80,000 per cm^2^ was used. For the 3D culture system, a cell density of 500,000 cells/mm^3^ was seeded. GCTB primary cultures were maintained in DMEM supplemented with 10% fetal bovine serum (Invitrogen), 1% penicillin/streptomycin and 1% glutamine at 37 °C in a 5% CO_2_ atmosphere. Replacement of medium was performed twice a week. Low-passage and actively proliferating primary cell cultures were used for all the experiments.

### 2.7. Establishment of GCTB and DF Patient-Derived Primary Cultures

Patient-derived primary cells were analysed for their histopathological features with hematoxylin and eosin (H&E) staining according to the manufacturer’s instructions. For patient-derived primary cultures, 100,000 cells were cytospun onto glass slides, fixed with acetone for 10′ and chloroform for 5′. For tumor tissue, 5 µm-thick slices were stained following the same protocols described above.

### 2.8. Building of Collagen-Based Scaffold 3D Culture Model

Synthesis of tridimensional collagen based-scaffold culture systems was performed in our laboratory as follows: in brief, a 1% wt suspension of bovine-derived insoluble type I microfibrillar collagen was dispersed in 0.05 M of acetic acid solution. Subsequently, 1 M of sodium hydroxide solution was added to the mixture which was then cross-linked with 1 wt% 1,4-butanediol diglycidyl ether (BDDGE) solution to stabilize the collagen matrix and to adjust porosity and tortuosity. The obtained suspension was homogenized with IKA T18 Basic ULTRA-TURRAX and centrifuged to remove air bubbles. The mixed solution was then freeze-dried with a controlled freezing and heating ramp under vacuum conditions in order to achieve optimal pore interconnectivity and to consolidate the matrix architecture. Scaffolds were then sterilized in ethanol 70% for 1 h and washed three times with PBS before use for in vitro analyses.

### 2.9. Chemobiogram Analysis

A 3-(4,5-dimethylthiazol-2-yl)-2,5-diphenyltetrazolium bromide (MTT) reduction assay was used to assess the efficacy of chemotherapy. In brief, tumor cells were seeded in both 96-well plates and 3D collagen-based scaffolds at a density of 80,000 cells/cm^2^ and were exposed to drugs 3 days after. The treatments were selected according to peak plasma concentration of each drug extrapolated from pharmacokinetic clinical data: lenvatinib (LENVA) 0.6 µg/mL (Eisai Ltd., Milan, Italy) [35], denosumab (DENO) 27 µg/mL (Amgen Inc., Milan, Italy) [36]. After 72 h of drug exposure, cells survival percentage was assessed. The experiments were performed twice.

### 2.10. Wound Healing Assay

Scratch wound assay was performed as follows: briefly, 2.5 × 10^5^ patient-derived primary cells were seeded using Culture-Insert 2 Well in µ-Dish 35 mm (Ibidi, Gräfelfing, Germany). After 24 h of cell attachment, the culture was exposed to DENO (Amgen Inc., Milan, Italy) LENVA (Eisai Ltd., Milan, Italy) and DENO + LENVA or vehicle for 72 h. Images were captured with EVOS XL Cell Imaging System (Thermo Fisher Scientific, Waltham, MA, USA) at 0 and 72 h and the cell migration rate was obtained by observing the wound closure after 72 h treatment exposure and compared to untreated cells.

### 2.11. Zebrafish Xenotransplant Model

Zebrafish husbandry procedures were performed according to the Directive 2010/63/EU and in compliance with local animal welfare regulations (authorization n. prot. 18311/2016; released by the “Comune di Meldola”, 9 November 2016). Ab and fli1a wild-type strain fertilized eggs were obtained and cultured according to previous works [29,33]. The embryos were anesthetized with 0.02% tricaine solution before any manipulation. For DF1 engraftment, the embryos were dechorionated at 48 h postfertilization (hpf). Patient-derived tumor cells were red labelled (CellTracker™ CM-DiI, Invitrogen) using a concentration of 2.5 × 10^5^/µL. From 300 to 500 cells were injected in the yolk sack or in perivitelline space of 48 hpf embryos. DF1-grafted embryos were treated with DENO (Amgen Inc., Milan, Italy) LENVA (Eisai Ltd., Milan, Italy) and DENO + LENVA, or not treated. Embryos were exposed to the drugs at 32 °C for 72 h and imaged through a fluorescence stereomicroscope (Nikon SMZ 25 equipped with NIS Elements software) at 2, 24 and 72 hpi. Untreated embryos were also imaged using an A1 laser confocal microscope (Nikon Corporation, Tokyo, Japan), and images were analyzed with the NIS Elements software (Nikon Corporation, Tokyo, Japan).

### 2.12. Statistical Analysis

Each experiment was performed in three independent replicates. Data are presented as mean ± standard deviation (SD), or mean ± standard error (SE), as reported. Two-tailed Student’s *t*-test was used to assess differences between groups, accepted as significant at *p* < 0.05.

## 3. Results

### 3.1. Patient Clinical History

GCTB1 was a 45-year-old female patient with a diagnosis of right proximal tibia GCTB. No relevant anamnesis was reported at the time of diagnosis. After the diagnosis of GCTB the patient received neoadjuvant denosumab 120 mg from May 2018 to October 2019 with partial response. During treatment, the patient reported osteonecrosis of the right inferior jaw (ONJ) which was treated with conservative surgery. In January 2019, the patient refused invasive surgical treatment with knee prosthesis, she underwent conservative curettage and cementation for GCTB of the right proximal tibia. Due to a minimal residual disease shown in the postsurgery computed tomography (CT) scan (Appendix A), the Multidisciplinary Board suggested indication to continue the treatment with denosumab. The patient moved to another country and was lost in follow-up.

GCTB2 was a 39-year-old female patient with a diagnosis of right distal tibia GCTB. In April 2019 there was the appearance of pain in the right ankle with no apparent trauma. X-ray and MRI evaluation of October and November 2020 showed distal tibial osteolysis with marked posterior extraosseous component (Appendix A). The images were compatible with GCTB lesions, but for a correct diagnosis a CT-guided biopsy was programmed. Histological analysis confirmed the diagnosis of GCTB. In November 2020 the patient underwent curettage and cementation. CT evaluation showed no evidence of local relapse. The patient is still on follow-up with no evidence of recurrent disease.

GCTB3 was a 43-year-old male with a diagnosis of right distal ulna GCTB. In January 2021, the patient was referred to our center after the appearance of pain. X-ray and MRI examinations were suggestive for a giant cell tumor of bone (Appendix A). The Multidisciplinary Board suggested indication for curettage and cement surgery after biopsy. After surgery, in October 2021, X-ray evaluation showed no evidence of diseases. The patient is still on follow-up.

GCTB4 was a 47-year-old male with a diagnosis of left ischium bone and posterior pilastrum of the acetabulum GCTB. In April 2014, MRI examinations were suggestive for a giant cell tumor of bone. The diagnosis was confirmed by CT-guided biopsy (Appendix A). The patient received neoadjuvant denosumab 120 mg from April 2014 to April 2017 with complete clinical, metabolic (18.9 to 2.1 SUV) and morphological response with peripheral calcification with no indication of surgery due to the extension of disease. He started denosumab again in September 2017 after the recurrence of pain and increased metabolic activity at PET: SUVmax = 5.03; previous equal to 2.1. Due to a good new response to the treatment with metabolic normalization, the patient received denosumab 120 mg every 3 months from September 2019 to April 2021. Since the FDG PET scan showed a new increase in metabolism in the lesion, patient started denosumab again at 120mg every 28 days. The patient is still on treatment with stable disease.

DF1 was a 24-year-old male. In July 2021, following abdominal pain, the patient performed an abdominal CT scan in suspicion of renal colic with occasional finding of lytic lesion of the left femoral neck (Appendix A). CT-guided biopsy was performed with initial diagnosis of possible GCTB. In August 2021, the patient underwent curettage and concreting. The definitive histological examination made the diagnosis of desmoplastic fibroma. The patient is currently in follow-up with no evidence of residual or disease recurrence.

### 3.2. Diagnosis of GCTB and DF Case Series

Macroscopic evaluation of surgically resected right proximal tibia tumor tissue of patient GCTB1 revealed numerous fragments of yellowish material with a total weight of 404 gr. Hematoxylin and eosin-stained tumor tissue (Figure 1A) reviewed by an experienced sarcoma pathologist revealed abundant fragments of giant cell tumor, areas of necrosis and fragments of newly formed bone tissue due to previous therapy. The diagnosis was GCTB.

Macroscopic evaluation of surgically resected right distal tibia tumor tissue of patient GCTB2 revealed various greyish fragments of tense-elastic consistency measuring 9 × 8 × 2 cm. Hematoxylin and eosin-stained tumor tissue (Figure 1A) reviewed by an experienced sarcoma pathologist revealed fragments consisting of mononuclear cells with rare mitosis and numerous giant cells. Necrosis and remodeled marginal bone trabeculae were observed. The diagnosis was GCTB.

Macroscopic evaluation of surgically resected right distal ulna tumor tissue of patient GCTB3 revealed 5 cm bone fragment including brownish nodular neoformation with 3 cm clear margins. Hematoxylin and eosin-stained tumor tissue (Figure 1A) reviewed by an experienced sarcoma pathologist revealed fragments of giant cell tumor. The diagnosis was GCTB.

Incisional sampling biopsy of left sinus ischium pubic branch tumor tissue of patient GCTB4 was performed. Hematoxylin and eosin-stained tumor tissue (Figure 1A) reviewed by an experienced sarcoma pathologist revealed compact proliferation sections of osteoclastic giant cells mixed with mononuclear elements with similar nuclear characteristics. Cell proliferation was covered by a fine capillary network. There were neither atypical elements in interstitial proliferation nor aspects of fibrosis or granulating tissue. Mononuclear cells were positive for the histiocyte marker CD163 and negative for protein S100. The diagnosis was GCTB.

Macroscopic evaluation of surgically resected left femur tumor tissue of patient DF1 revealed brownish tissue fragments measuring 4.5 × 2 × 1 cm. Hematoxylin and eosin-stained tumor tissue (Figure 1A) reviewed by an experienced sarcoma pathologist revealed fragments of desmoplastic fibroma. The diagnosis was desmoplastic fibroma (DF).

Clinicopathological characteristics of GCTB and DF case series are summarized in Table 1.

### 3.3. Establishment of GCTB and DF Patient-Derived Primary Culture

GCTB1. Hematoxylin and eosin-stained primary cells reviewed by an experienced sarcoma pathologist revealed mononuclear and giant neoplastic cells with a proportion of 25% (Figure 1B). The establishment of GCTB1 patient-derived primary culture was confirmed.

GCTB2. Hematoxylin and eosin-stained primary cells reviewed by an experienced sarcoma pathologist revealed mononuclear cells with a proportion of 70% (Figure 1B). The establishment of GCTB2 patient-derived primary culture was confirmed.

GCTB3. Hematoxylin and eosin-stained primary cells reviewed by an experienced sarcoma pathologist revealed mononuclear and giant neoplastic cells with a proportion of 50% (Figure 1B). The establishment of GCTB3 patient-derived primary culture was confirmed.

DF1. Hematoxylin and eosin-stained primary cells reviewed by an experienced sarcoma pathologist revealed mononuclear cells with a proportion of 35% (Figure 1B). The establishment of DF1 patient-derived primary culture was confirmed.

### 3.4. Gene Expression Profiling in GCTB and DF Cases

In order to provide evidence of the role of new molecular targets and to support the use of bone-targeted therapy and tyrosine kinase inhibitors in GCTB, a panel of bone-related markers and epithelial-to-mesenchymal transition (EMT)- and angiogenesis-related markers were investigated (Figure 2*)*. In this regard, the expression of the axis *RANKL*/*RANK*/*OPG*—well-known markers involved in bone vicious cycle, osteoclastogenesis and the first molecular target of the monoclonal antibody denosumab—was modulated in respect to controls. In particular, *RANKL* was significantly upregulated in all the patient-derived primary cultures with respect to control. *RANK* was significantly upregulated in GCTB2 and GCTB3 but not in DF1 and the patient-derived primary cultures with respect to control. *OPG* was upregulated in GCTB2 and downregulated in GCTB1, GCTB3 and DF1 with respect to control. Moreover, the other bone-related marker *CXCR4*, a master regulator of cell migration, hematopoiesis, cell homing and retention in bone marrow [37], was significantly upregulated in all analyzed cases. Furthermore, *CXCL12*—a chemokine 12 able to bind to its cognate receptor *CXCR4* determining the activation of several downstream signaling pathways which regulate tumor progression and metastasis—was found to be significantly downregulated in all the case series. *OPN*, a bone matrix protein which orchestrates biological events involving the immune system and the vascular system [38], was significantly upregulated in all the patient-derived primary cultures. *RUNX2*, a key molecule in the transformation of bone marrow mesenchymal stem cells to osteoblasts, which is also involved in tumor invasion and metastasis, was significantly upregulated in all analyzed cases. Finally, *FLT1*, a maker involved in several endothelial cell pathways including cell proliferation, migration and vascular permeability [39], was downregulated in GCTB1 and upregulated in all the other investigated patients. All fold-change values and related p values are reported in Appendix A.

### 3.5. Bone-Targeted Therapy and Tyrosine Kinase Inhibitor Assessment in monolayer and Tridimensional GCTB and DF Primary Culture Model Case Series

We aimed to investigate the activity of bone-targeted therapy and multireceptor tyrosine kinase inhibitor in monoregimen or in combination in GCTB. Thus, we exposed our GCTB patient-derived primary culture case series, both cultured in 2D and within 3D scaffolds, to DENO, to LENVA and to the combination of DENO and LENVA.

The results showed the tumor sensitivity to the tested drugs. In particular, across all patients the most active treatment was the combination of DENO and LENVA both in 2D and 3D (Figure 3). Moreover, LENVA exhibited higher activity both in 2D and 3D compared to bone-target therapy DENO, which represents one of the recommended drugs in the neoadjuvant setting for patients with advanced GCTB who were not candidates for primary curettage [40]. Taking in consideration DENO, cancer cells’ survival significantly decreased with respect to CTR only in GCTB2, while no significant differences were detected in GCTB3 and DF1 (Figure 3B,C,E,F). Differences in GCTB primary cultures’ confluence and morphology were observed in 2D cultures, confirming the higher sensitivity to the combination of DENO and LENVA (Figure 3D–F) and in 3D purple formazan crystals formation (Figure 3E,F).

### 3.6. The Impact of Bone-Targeted Therapy and Tyrosine Kinase Inhibitor Combination on GCTB and DF Primary Cell Culture Migration

The activity of bone-targeted therapy and tyrosine kinase inhibitor on cell migration was assessed both in GCTB3 and DF1. As shown in Figure 4, in the GCTB3 untreated primary cells, the wound completely disappeared after 72 h. DENO exhibited a 14%, LENVA 50% and DENO + LENVA 62% cell-free surface area compared to the respective baseline. Similarly, in the DF1 untreated primary cells, the wound also completely disappeared after 72 h. DENO exhibited a 12%, LENVA 78% and DENO + LENVA 80% cell-free surface area compared to the respective baseline. These data indicate that these molecules exerted antitumor activity involving the inhibition of cells’ motility.

### 3.7. The Role of Bone-Targeted Therapy and Tyrosine Kinase Inhibitor in DF Primary Culture Xenotransplanted Zebrafish Model

In order to explore the activity of bone-targeted therapy and tyrosine kinase inhibitor combination in DF bone lesion, DF1 was xenotransplanted in zebrafish embryos. The engraftment was successfully achieved in both yolk sack and perivitelline duct (Figure 5 and Appendix A) and the tumor growth was monitored at 2, 24 and 72 h postinjection (hpi). Tumor-growth imaging revealed equivalent fluorescence signals at 2 hpi in all tested conditions. A decreased fluorescence signal was observed with LENVA and DENO + LENVA compared to both DENO and untreated embryos at 24 hpi (Figure 5). Furthermore, the tumor-growth imaging detected at 72 hpi showed a decrease in fluorescence signal with LENVA compared to both DENO and untreated embryos, while no embryos survived with DENO + LENVA (Figure 5). In this regard, among all tested conditions the mortality was 10% in untreated embryos, 25% in DENO group, 44% in LENVA group and 100% with DENO + LENVA, while no severe abnormalities were detected across the tested conditions (Appendix A). The results are suggestive of the activity of bone-targeted therapy and tyrosine kinase inhibitor in DF.

## 4. Discussion

In this work, we aimed to explore the role of new molecular targets and treatments for poorly explored bone sarcoma histotypes, including lesions which are considered benign but exhibit a locally aggressive behaviour, such as GCTB and DF. For the above reasons, we conceived a prospective study taking advantage of our transnational platform combining the use of patient-derived primary culture, 3D scaffolds and zebrafish model. Six patients were included in the study, five of them affected by GCTB and one by DF; the enrollment started in January 2019 and ended in August 2021.

First, we isolated GCTB and DF primary cells from the surgical specimens and we achieved the establishment of primary cultures (Figure 1). Next, in order to deepen the biology of these less understood diseases and to provide the rationale for testing new chemotherapeutics, we analyzed a panel of bone- and neoangiogenesis-related markers which are known to be involved or targeted by bone target therapy and antiangiogenic drugs, including tyrosine kinase inhibitors.

In particular, the axis *RANK*/*RANKL*/*OPG*—which is responsible for orchestrating the bone vicious cycle and the activation of osteoclastogenesis potential—was significantly unbalanced towards the bone resorption activation, with the upregulation of *RANK* and *RANKL* and the downregulation of *OPG* in all the tested samples compared to control (Figure 2, Appendix A). These results provide a rationale for testing bone-targeted therapy, such as denosumab, in these specific histotypes, and are consistent with the clinical evidence that denosumab is effective in GCTB. Interestingly, although a significant upregulation of *RANKL* was observed in DF1, a slight upregulation of *RANK* was detected, supporting the evidence of a limited role of denosumab in this specific histotype.

Moreover, the observed results were further corroborated by the expression analysis of other bone master regulators *RUNX2* and *CXCR4*, which were found significantly upregulated. Furthermore, the analysis of genes involved in bone vascular formation, such as *OPG,* and in neoangiogenesis, such as *FLT1,* showed a significant downregulation of the first and the upregulation of the latter. These results are suggestive of a potential role of new antiangiogenic generation drugs, including the multitarget TKI inhibitor lenvatinib, as promising drugs for these lesions (Figure 2). Finally, a downregulation of *CXCL12* was observed.

The above observations prompted us to test bone resorption inhibitor and antiangiogenic drugs in these specific bone sarcoma histotypes.

In this regard, the human monoclonal antibody denosumab, is a Receptor Activator of Nuclear factor κB Ligand (RANKL) inhibitor [41]. Impairing the activation of RANK ligands leads to the inhibition of osteoclasts’ activation and to the reduction in tumor-associated bone lysis [42]. GCT are characterized by both osteoclast-like giant cells expressing the Receptor Activator of Nuclear factor κB (RANK) and stromal cells expressing RANK ligand. The so-called vicious cycle, which is established between these two cellular populations, is responsible for bone resorption in bone sarcomas and metastases. Currently, denosumab is approved for the treatment of GCT, showing efficacy in unresectable/metastatic disease as well as in the neoadjuvant setting [43].

Considering multitargeted tyrosine kinase inhibitors, such as lenvatinib, they act through the inhibition of vascular endothelial growth factor receptors (VEGFR) including VEGFR1 (FLT1), VEGFR2 (KDR) and VEGFR3 (FLT4). Moreover, they impair the activity of proangiogenic and pro-oncogenic tyrosine kinase receptors such as fibroblast growth factors (FGF) FGFR1, 2, 3 and 4, platelet-derived growth factor receptor A (PDGFRα), proto-oncogene receptor tyrosine kinase (KIT) and rearranged during transfection (RET) kinase. In this regard, lenvatinib is currently approved or under investigation for the treatment of many solid tumors [44]. Its efficacy is under evaluation in the treatment of selected metastatic and/or unresectable STS and bone sarcomas including osteosarcoma and chondrosarcoma (NCT04784247) in combination with pembrolizumab. Angiogenesis is one of the major hallmarks of tumorigenesis, and the involvement of VEGFR has been described in supporting RANKL-induced osteoclastogenesis in GCTB [45,46].

Thus, we exposed our patient-derived primary GCTB and DF case series, both cultured with standard monolayer culture and within 3D scaffolds, to DENO, LENVA and DENO+LENVA (Figure 3). The results showed that DENO exerted cytotoxic activity in GCTB2, while it did not affect cell survival in GCTB3 and DF1. In this regard, although the expression of *RANKL* was upregulated both in GCTB3 and DF1, the slight upregulation of *RANK* in DF1 could in part explain the lower sensitivity to DENO observed in this culture. Interestingly, the cell culture sensitivity to DENO was expected to also be high in GCTB3, as observed in GCTB2, while this was not seen either in 2D or 3D culture. These data could be linked to the lower expression of *RANKL, CXCR4* and *RUNX2,* which could have affected the sensitivity to DENO compared to GCTB2. In this regard, the observed results suggest a potential role of these biomarkers in predicting response to bone target therapy, however further analyses are needed in order to confirm this hypothesis. Furthermore, DENO did not affect the viability of DF1 cells, which could be explained by the lower expression of *RANK* compared to GCTB2 and GCTB3.

Considering the multitarget tyrosine-kinase inhibitor (TKI) LENVA, the results showed a higher sensitivity to this drug across all primary cultures, both in 2D and 3D compared to DENO. These results are suggestive of the promising role of this new antiangiogenic generation drug in the management of these lesions. In this regard, a recent multicentre, open-label, multicohort, phase 1/2 trial has highlighted lenvatinib as a promising antitumor drug in other bone sarcoma histotypes, such as osteosarcoma, in combination with etoposide plus ifosfamide. The results showed an enhanced efficacy of chemotherapy mediated by lenvatinib, with no new safety issues in patients [47]. Furthermore, ongoing trials (NCT04784247] are assessing the role of lenvatinib and pembrolizumab in patients with advanced soft tissue sarcoma (STS) including, among all, some of the vascular sarcomas and bone sarcomas. Finally, in vitro pharmacological profiling and tumor migration assay showed that the combination of DENO and LENVA exerted a higher cytotoxic activity across all primary cultures, providing encouraging preliminary data for deepening the role of this combination in the clinical setting.

Ex vivo observations were further validated through the use of xenotransplanted DF1 in zebrafish embryos (Figure 5). In this regard, in vivo pharmacological profiling corroborated the role of bone resorption inhibitors and antiangiogenic drugs in less explored DF1 lesions, especially with LENVA and DENO + LENVA at 24 hpi. The obtained results at 72 hpi showed a decreased fluorescence signal with LENVA compared to both DENO and untreated embryos, while no embryos survived with DENO + LENVA. The latter results suggested a toxicity profile of the combination. Indeed, it should be taken into consideration that, in order to achieve a translationally relevant result, the human plasma peak concentrations of these drugs were used, and this may have affected the embryos’ survival. However, further analyses are needed to confirm these data.

Nowadays, anecdotal information about the role of chemotherapy is reported for DF. In this regard, although DF represents an extremely rare benign bone tumour (0.11% of all primary bone tumours [48]), it exhibits a locally aggressive behaviour. Clinical features reported in the latest WHO described DF patients with longstanding history of pain or bone deformity. Moreover, 10% of cases presented a pathological fracture. Chemotherapy agents, such as vincristine, doxorubicin and dacarbazine (DTIC) have been used with limited activity [26]. Thus, there is an urgent clinical need to find new therapeutic strategies for these patients. The above considerations prompted us to investigate new targets and drugs in these neoplasms.

We are aware that our study is not free from limitations, mainly represented by the small number of patients enrolled and the heterogeneity in primary culture tumor cells. Moreover, the use of preclinical models represents another criticism due to the gap which persists between clinical and translational observations. In order to partially solve some of these drawbacks, we combined the use of different preclinical platforms including 3D culture systems, the use of patient-derived primary cultures, and zebrafish in vivo model and clinical data.

To the best of our knowledge, this translational study represents the first in the literature in which GCTB and DF biology and response to anticancer drugs have been explored through the use of patient-derived primary cultures, 3D culture system and a zebrafish model. Furthermore, to date, approximately 200 cases of DF have been reported in the literature with only 8 cases occurring in the proximal femur. Therefore, our case represents the ninth DF described in literature reporting an occurrence in the proximal femur.

Taken together, our results are consistent with the clinical observation that denosumab is effective in GCTB [49,50]. Moreover, our results are suggestive of a promising role of Lenvatinib, in monoregimen or in combination, for the management of GCTB and DF, providing some evidence for better exploring the use of these drugs in these poorly understood diseases. Finally, these preliminary results shed light on the tumor biology of these lesions, highlighting the promising role of some biomarkers as potential predictive and druggable targets for these diseases. Further research is needed in order to further confirm this evidence.

## Figures and Tables

**Figure 1 biomedicines-10-00372-f001:**
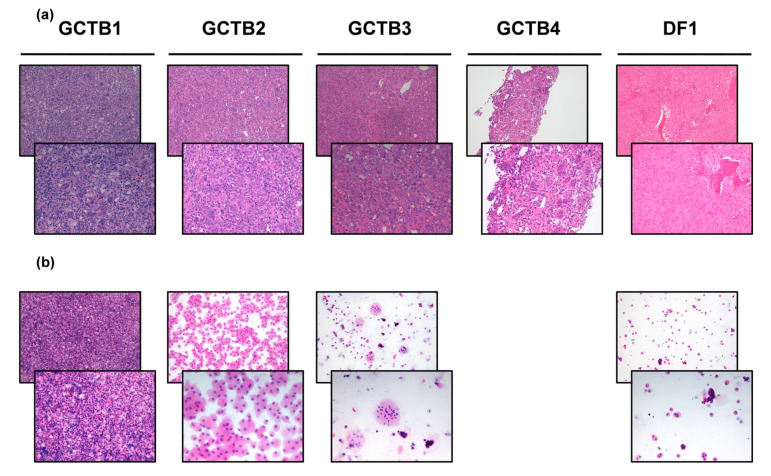
(**a**) H&E of the patient’s surgically resected tumor specimen (10× and 20× magnification). (**b**) H&E of the patient-derived primary culture (10× and 20× magnification).

**Figure 2 biomedicines-10-00372-f002:**
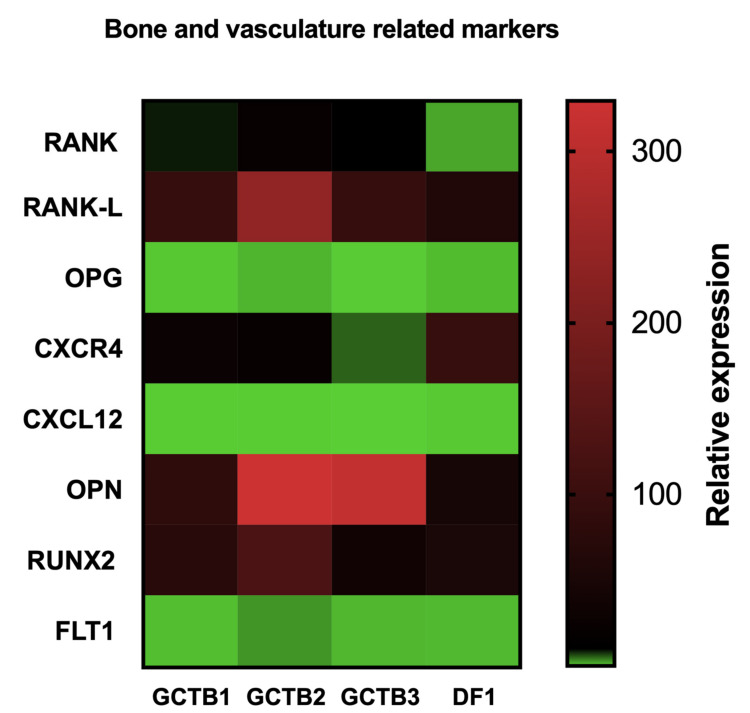
Heat map comparison of the relative gene expression of bone- and vasculature-related markers in GCTB and DF tumor tissue compared to the matched healthy tissue.

**Figure 3 biomedicines-10-00372-f003:**
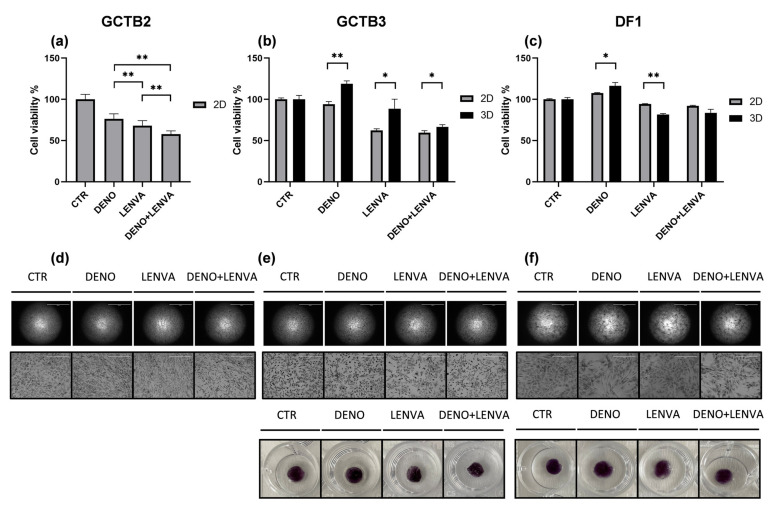
Pharmacological analysis of: (**a**) 2D GCTB2 primary culture, (**b**) 2D and 3D GCTB3 primary culture, (**c**) 2D and 3D DF1 primary culture. Representative images of (**d**) 2D GCTB2, (**e**) 2D and 3D GCTB3 (**f**) 2D and 3D DF1, exposed to the tested drugs, 2× and 10× magnification. GCTB and DF primary cultures were exposed to DENO, LENVA, DENO + LENVA. Significant differences among treatments were accepted for *p* < 0.05 (*) and *p* < 0.001 (**).

**Figure 4 biomedicines-10-00372-f004:**
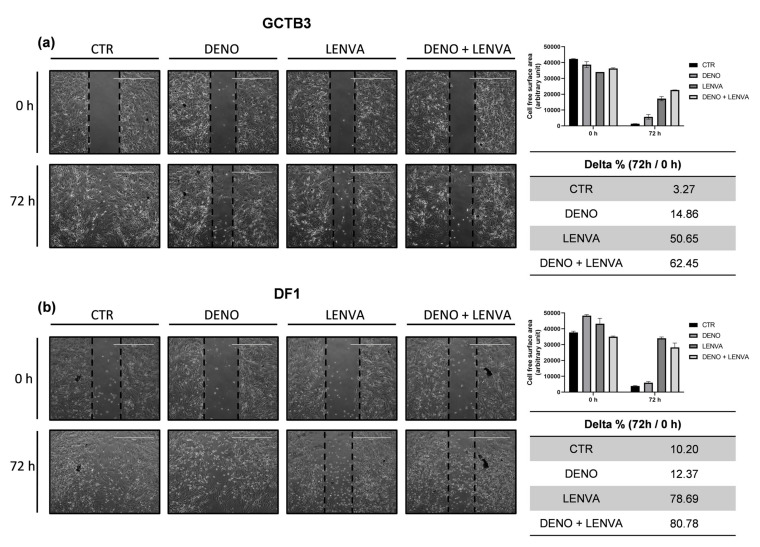
Migration assay analysis. (**a**) Wound closure was assessed after 72 h of drug exposure or without treatment, representative 4× images of GCTB3 primary culture (control sample and treated). (**b**) Representative 4× images of DF1 primary culture (control sample and treated). Scale bar 1000 µm.

**Figure 5 biomedicines-10-00372-f005:**
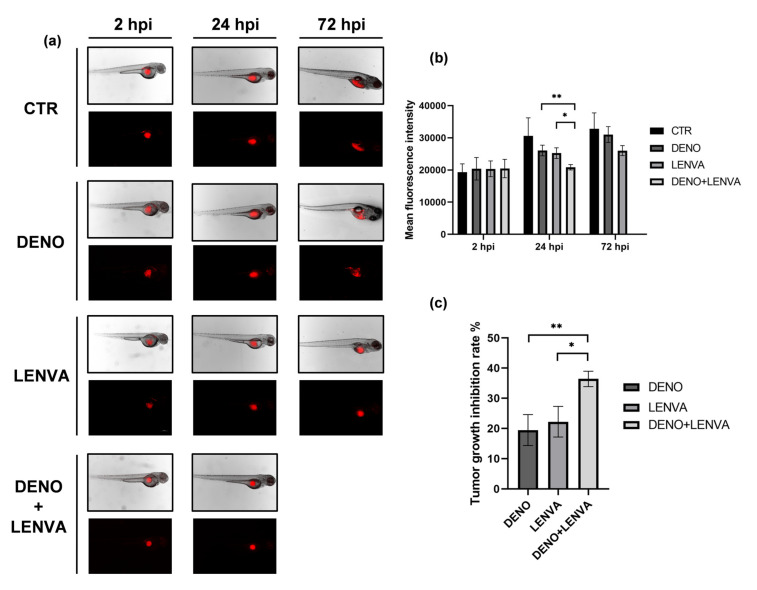
(**a**) Representative fluorescence microscopy images of zebrafish embryos xenotransplanted with DF1. Images of embryos untreated and exposed to DENO, LENVA and DENO + LENVA at 2, 48 and 72 hpi, scale bar 1000 μm. (**b**) Mean fluorescence signal of DF1 xenotransplanted embryos, arbitrary units. (**c**) Tumor-growth inhibition rate between tested drugs. Significant differences among treatments were accepted for *p* < 0.05 (*) and *p* < 0.001 (**).

**Table 1 biomedicines-10-00372-t001:** Clinicopathological characteristics of GCTB and DF case series.

Patient	Gender	Age at Surgery	Site	Size (cm)	Histological Subtype	IHC Analysis	Surgical Margins	Radiotherapy Postsurgery	Chemotherapy Pre/Post Surgery	Follow-Up Months
GCTB1	F	45	right proximal tibia		Abundant fragments of giant cell tumor, areas of necrosis and fragments of newly formed bone tissue	na	R1	na	Neoadjuvant Denosumab 120 mg with PR and right inferior mandibular ONJ	42
GCTB2	F	39	right distal tibia	9 × 8 × 2	Fragments consisting of mononuclear cells with rare mitosis and numerous giant cells. Necrosis and remodeled marginal bone trabeculae were observed	na	R0	na	na	19
GCTB3	M	43	right distal ulna	5 × 3 × 1	Fragments of giant cell tumor	na	R0	na	na	10
GCTB4	M	47	left sciatic bone and posterior pillar of the acetabulum		Compact proliferation sections of osteoclastic giant cells mixed with mononuclear elements with similar nuclear characteristics. Cell proliferation was covered by a fine capillary network. There were no atypical elements in interstitial proliferation nor aspects of fibrosis or granulating tissue	CD163 +S100 -	na	na	Adjuvant denosumab 120 mg with CR	90
DF1	M	24	left femoral head and neck	4.5 × 2 × 1	fragments of desmoplastic fibroma	na	R0	na	na	2

Giant cell tumors, GCT; M, male; F, female; PR, partial response; CR, complete response; ONJ, osteonecrosis of the jaw.

## Data Availability

The datasets generated and/or analyzed during the current study are available from the corresponding author on reasonable request.

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
