# Peer review of "A Rationale for the Activity of Bone Target Therapy and Tyrosine Kinase Inhibitor Combination in Giant Cell Tumor of Bone and Desmoplastic Fibroma: Translational Evidences"

_biomedicines, 2022, doi:10.3390/biomedicines10020372_

Round 1
Reviewer 1 Report
The manuscript entitled “A rationale for the activity of bone target therapy and tyrosine kinase inhibitor combination in giant cell tumor of bone and desmoplastic fibroma: translational evidences” describes that giant cell tumor of bone (GCTB) and desmoplastic fibroma (DF) are bone sarcomas with intermediate malignant behavior and unpredictable prognosis. These locally aggressive neoplasms exhibit a predilection for the long bone or mandible of young adult, causing a severe bone resorption. In particular, the tumor stromal cells of these lesions are responsible for the recruiting of multinucleated giant cells which ultimately lead to bone disruption. In this regard, the underlying pathological mechanism of osteoclastogenesis process in GCTB and DF is still poorly understood. In this study, authors report for the first time a molecular and pharmacological investigation of GCTB and DF combining the use of translational and clinical data.
- What are the role of genes including the upregulation of RANK-L, RANK, OPN, CXCR4, RUNX2 and the downregulation of OPG, CXCL12 and FTL1 in this study?
- In the figure 3, the results from 2D and 3D studies are not consistent.
- In the figure 4, authors should use transwell to measure cellular migration. Wound closure is influenced by cell growth/proliferation rate. High cell growth or proliferation rate will has well wound closure.
- In the figure 4, authors should use GCTB in this study.
Author Response
-What are the roles of genes including the upregulation of RANK-L, RANK, OPN, CXCR4, RUNX2 and the downregulation of OPG, CXCL12 and FTL1 in this study?
As mentioned in the Results and Discussion sections, these genes are involved in bone vicious cycle and osteoclastogenesis (RANK, RANKL, OPG), in neoangiogenes (FTL1), in bone vascular formation (OPG), in osteoblast maturation (RUNX2, OPN), in migration of tumor cells from bone into circulation and for developing systemic disease and metastases (CXCR4, CXCL12), and are targets of the tested drugs. We also added a Graphical abstract depicting the investigated pathway.
-In the figure 3, the results from 2D and 3D studies are not consistent.
The establishment of a primary cell culture from a surgically-resected tumor tissue is rarely a simple procedure. Moreover, the number of research experiments that can be carried out on primary tissue specimens is often limited, mainly because of the small amount of material available from these samples and also because of the number of steps initially required to isolate cells suitable for in vitro assays. In our manuscript we successfully obtained the establishment of GCTB and DF primary cultures, but the amount of GCTB2 isolated tumor cells did not allow us to perform pharmacological analysis through the 3D culture system. Due to the valuable and translational results obtained from patient-derived primary cultures we included these data in the manuscript, although a comparison of the pharmacological profile of GCTB2 is possible only to 2D profiles obtained with GCTB3 and DF1, and not with 3D. Moreover, the expected decrease in drugs activity using 3D platforms is not always observed as shown for LENVA condition in DF1 culture. This is explainable, as already demonstrated by our group (Liverani et al, Biol. Open. 2017), by the ability of our model to increase the expression of some STS associated-markers and to select a higher tumor cells percentage compared to standard 2D cultures. This could explain the increase in sensitivity observed in 3D in some cases. However, the heterogeneity of primary cultures tumor cells, the rarity of the investigated disease, the limited reliability of the preclinical models in reproducing the full spectrum of all tumor features represent study limitations as reported in the Discussion section.
-In the figure 4, authors should use transwell to measure cellular migration. Wound closure is influenced by cell growth/proliferation rate. High cell growth or proliferation rate will has well wound closure.
Thank you for your suggestion. We previously characterized the proliferation rate of the tested primary cultures before performing the scratch wound assay. In this regard, the doubling time was 86 h for GCTB3 and 94 h for DF1. Thus, we are confident that the observed results were related to cellular migration and not influenced by cell growth/proliferation rate. Of course, we will take in mind your suggestions for future characterizations.
-In the figure 4, authors should use GCTB in this study.
As shown in the upper panel of Fig. 4 a), we performed the migration assay using GCTB3 as well, reporting representative images of wound healing and quantification of cell free surface area after 72h of treatment.
Reviewer 2 Report
This paper by De Vita et al. provides initial data on the impact to GCTB cancer cells from a dual DENO/LEVA treatment strategy. As stated in the paper title, the abstract, and the introduction sections, this study's purpose is to provide translational evidence for the rationale of treating GCTB with both a tyrosine kinase inhibitor and an inhibitor of bone turnover.
Regarding the data presented in the paper: the combination of the two inhibitors provide only modest results compared to monotherapy. While some minor statistical significance was achieved, an honest evaluation of the data would be hard pressed to conclude that biological significance was achieved in the cell models. However, the experiments were performed according to acceptable standards and the reported data will be of interest to the field.
However, as mentioned above, the title, abstract, introduction, and conclusion sections all indicate that this study is to provide evidence of a dual treatment rationale/strategy. However, the authors completely fail to provide any context whatsoever about the mechanisms of action for either drug. For example, the term "tyrosine kinase inhibitor" is in the title of the paper but the authors do not spend any effort to describe either why inhibiting tyrosine kinase activity might be a suitable rationale for treating GCTB or which drug inhibits TK activity or how/which TK activities are inhibited. The same is true for the other drug. Biomedicines is a journal with broad readership and therefore the authors must significantly improve this portion of their manuscript.
Author Response
Regarding the data presented in the paper: the combination of the two inhibitors provide only modest results compared to monotherapy. While some minor statistical significance was achieved, an honest evaluation of the data would be hard pressed to conclude that biological significance was achieved in the cell models. However, the experiments were performed according to acceptable standards and the reported data will be of interest to the field. However, as mentioned above, the title, abstract, introduction, and conclusion sections all indicate that this study is to provide evidence of a dual treatment rationale/strategy. However, the authors completely fail to provide any context whatsoever about the mechanisms of action for either drug. For example, the term "tyrosine kinase inhibitor" is in the title of the paper but the authors do not spend any effort to describe either why inhibiting tyrosine kinase activity might be a suitable rationale for treating GCTB or which drug inhibits TK activity or how/which TK activities are inhibited. The same is true for the other drug. Biomedicines is a journal with broad readership and therefore the authors must significantly improve this portion of their manuscript.
We thank you for your valuable comments. We modified the paper according to your suggestions using a softer approach throughout the manuscript. Moreover, in the Discussion section we provided more detailed explanations about the mechanism of action of the tested drugs and their potential synergistic activity through the inhibition of the bone vicious cycle and the suppression of angiogenesis related pathways including the use of a graphical abstract.
Reviewer 3 Report
The paper is interesting.
Some suggestions/corrections:
- Ln 108: “(19)- can” is not correct. Please check the right general insert of “-“ in all the text.
- In Results a reference of “Supplementary Figure 1” is missing.
- In Supplementary Figure 1 there are RX and CT scan images.
- In caption of table 1 definition of ONJ is missing.
- Ln 396 “EMT” definition is missing.
- Ln 444: “case serie” is not correct.
- The start and the end of enrolment of patients should be inserted in “2.1. Ethical statement and case series”.
- Ln 564: definition of STS is missing.
- Ln 584: “Iongstanding”, with I and not L as initial letter, is not correct.
- Ln 599: “was explored” is not correct.
Author Response
The paper is interesting.
Some suggestions/corrections:
-Ln 108: “(19)- can” is not correct. Please check the right general insert of “-“ in all the text.
We corrected this issue throughout the manuscript.
-In Results a reference of “Supplementary Figure 1” is missing.
We added reference to Supplementary Figure 1 in the Result section of the manuscript.
-In Supplementary Figure 1 there are RX and CT scan images.
We modified the caption of Supplementary Figure 1 as kindly suggested.
-In caption of table 1 definition of ONJ is missing.
We added the definition of ONJ in the caption of Table 1 as kindly suggested.
-Ln 396 “EMT” definition is missing.
We added the definition of EMT in the appropriate section as kindly suggested.
-Ln 444: “case serie” is not correct.
We modified the terms in the appropriate section as kindly suggested.
-The start and the end of enrolment of patients should be inserted in “2.1. Ethical statement and case series”.
The information about start and end of enrollment was included at the beginning of the Discussion section. However, we added this information also in the Methods section as kindly suggested.
-Ln 564: definition of STS is missing.
We added the definition of STS in the appropriate section as kindly suggested.
-Ln 584: “Iongstanding”, with I and not L as initial letter, is not correct.
We modified the term in the appropriate section as kindly suggested.
-Ln 599: “was explored” is not correct.
We modified the terms in the appropriate section as kindly suggested.
Reviewer 4 Report
Authors introduce GCTB and DF extensively and investigate chemotherapy (den and lenva) in-vitro and in zebrafish. Outcome measures include tumor growth inhibition rate, expression of a set of target genes, cell migration (scratch) assay and cell viability. Whilst potentially interesting, the way the data is presented makes it hard to appreciate the results.
The abstract and introduction are well written. They could benefit from spacing between paragraphs. Additionally, gene / protein and other abbreviations should be written in full the first time they are mentioned. Moreover, the readability of the introduction would benefit from an overview figure of the pathway under investigation, visualize where the chemotherapie interacts with this pathway and visualize why the target genes were selected.
method: given the very detailed patient descriptions i think patients are pseudo anonymized (but I could be wrong)
Major comments
Please do not cite so many manuscripts in the method section. Describe the methods here in detail.
Please specify in more detail what is "related healthy tissue" and how it "relates" to the two different types of tumor.
Although nice a an overview, figure 1, 3 and 4 are way to small and of too low resolution to conclude anything from them. Please enlarge, increase resolution and explain (and point) in the figure legend to what the reader should look at /appreciate from them.
Figure 2: in the pharmacological analysis of 2d and 3d cultures cell viability is measured. It decreases upon exposure. Here, controls seem missing. Please elaborate on how to distinguish between a toxic effect (achievable with many chemicals) and a selected /higher toxic effect on tumors. Additionally, there is no mention of concentrations used. Are dose response relationships obtained prior to these experiments. Why are the concentrations used chosen? 3D culture :on the images provided, i do not see any difference between cultures.
paragraph 3.4 is unreadable due to the many abbreviations, foldchanges, p values tec. This information would really be suited for a table. A p-value of 0.05 would not be considered significant if multiple testing correction was applied (there are 8 measuring points / genes). Is there are correction for multiple testing? Upregulated /downregulated foldchanges would benefit from more information: up/down regulated compared to what value. (e.g. 0.0000003 compared to 0.00002 is extremely lower but the absolute value is close to 0 in both instances. What were the Ct values?
Figure 5: Visually there is hardly any difference between the intensities of the xenograft signal in these images.
Author Response
The abstract and introduction are well written. They could benefit from spacing between paragraphs. Additionally, gene / protein and other abbreviations should be written in full the first time they are mentioned.
We added the full name of genes, proteins and acronyms throughout the manuscript as kindly suggested.
Moreover, the readability of the introduction would benefit from an overview figure of the pathway under investigation, visualize where the chemotherapie interacts with this pathway and visualize why the target genes were selected.
Thank you for the suggestion. We added an appropriate Graphical abstract showing the pathways investigated and targets of selected drugs as kindly suggested.
method: given the very detailed patient descriptions i think patients are pseudo anonymized (but I could be wrong)
Patient anamnesis and description were reviewed by a Multidisciplinary Board, composed by a team of expert pathologists and clinicians in the field, which confirmed full anonymization of patients.
Major comments
Please do not cite so many manuscripts in the method section. Describe the methods here in detail.
Following Editor recommendations, we need to cite previously published manuscripts reporting methods details to reduce the risk that plagiarism is detected. However, we removed 5 references from the Methods section as kindly suggested.
Please specify in more detail what is "related healthy tissue" and how it "relates" to the two different types of tumor.
We thank Reviewer 4 for the suggestion. We now specified in the manuscript that “related healthy tissue” is the tissue coming from the margin of surgical excision for each tumor sample analyzed.
Although nice as an overview, figure 1, 3 and 4 are way too small and of too low resolution to conclude anything from them. Please enlarge, increase resolution and explain (and point) in the figure legend to what the reader should look at /appreciate from them.
We thank Reviewer 4 for the suggestion. Unfortunately, poor resolution of images depends on the software of Journal which compresses pictures when creating the file for peer review. We provided single original pictures with appropriate dimensions and high resolution.
Figure 2: in the pharmacological analysis of 2d and 3d cultures cell viability is measured. It decreases upon exposure. Here, controls seem missing. Please elaborate on how to distinguish between a toxic effect (achievable with many chemicals) and a selected /higher toxic effect on tumors. Additionally, there is no mention of concentrations used. Are dose response relationships obtained prior to these experiments. Why are the concentrations used chosen? 3D culture :on the images provided, i do not see any difference between cultures.
Negative controls are reported for each primary culture and culture system, both in the bar graphs and in the representative images. We added in the Discussion section the explanation of the rationale of testing these drugs due to specific target expression in the investigated primary tumor cells. This rationale was further corroborated by transcriptomic profiling data. For the above reasons, and as previously confirmed by our group, the activity of denosumab and lenvatinib is selective for tumor cells (for denosumab, please refer to: Spadazzi et al., J Bone Oncol 2019, Liverani et al., Bone 2014, Mercatali et al., Int J Mol Sci. 2016; for levatinib please refer to: De Vita et al., Int J Mol Sci. 2021 and De Vita et al., J Exp Clin Cancer Res 2021).
Moreover, in order to produce translational results, we tested each drug at the human plasma peak concentration obtained from clinical studies with the use of actively proliferating GCTB and DF primary cultures, as reported in the Methods section and in previous publications (De Vita et al., Ther Adv Med Oncol. 2017, De Vita et al., J Vis Exp. 2018, Miserocchi et al., Cells. 2018, De Vita et al, J Exp Clin Cancer Res. 2021, De Vita et al., Int J Mol Sci. 2017, De Vita et al., Molecules. 2016).
For the low images resolution, this depends on the manuscript template Journal policy, because of which the figures are pasted automatically in a Word file by the system. In order to provide higher resolution, we also separately uploaded each figure in tiff format. For 3D images, a decrease in formazan crystal is visible especially at the edge of the 3D culture systems upon exposure of drugs (less dark coloration if compared to controls).
paragraph 3.4 is unreadable due to the many abbreviations, foldchanges, p values tec. This information would really be suited for a table.
We thank Reviewer 4 for the suggestion. We removed FC and p value descriptions from Paragraph 3.4 and added Supplementary Table 1 reporting all relevant values.
A p-value of 0.05 would not be considered significant if multiple testing correction was applied (there are 8 measuring points / genes). Is there any correction for multiple testing?
As mentioned in the Methods section, differences between groups were assessed by a two-tailed Student’s t-test, and accepted as significant at p<0.05. No correction for multiple testing was performed since genes’ expressions were not compared among them. For each single gene, significance of differential expression between the tumor-derived primary culture and the healthy tissue from surgical margin was assessed.
Upregulated /downregulated foldchanges would benefit from more information: up/down regulated compared to what value. (e.g. 0.0000003 compared to 0.00002 is extremely lower but the absolute value is close to 0 in both instances.
Fold change of a given target gene is calculated as the difference between its expression in the tumor-derived primary culture sample compared to the related healthy tissue, which is the “control” value. In more detail, as stated in the Methods section, the relative expression ratio (fold change, FC) was calculated using 2-∆∆CT method. ∆CT were calculated subtracting the CT of the housekeeping gene to the CT of the target one, both for "test" (in this case, tumor-derived primary culture) and "control" (healthy tissue from surgical margin). Then, ∆∆CT was calculated.
What were the Ct values?
Ct values alone are not informative per se, since they are affected by initial RNA integrity, amplicon length, reverse transcription and amplification efficiency, primer specificity, ecc.. which could differ for each single sample. This is why typically FC are provided, since this value takes into account a normalization step against the housekeeping gene, whose expression is stable across different groups (healthy tissue and tumor samples), therefore its variation is virtually only due to all the variables mentioned above which are the very same affecting the target gene as well, so that the difference between the two (housekeeping and target gene) is “real”, effectively depending only to the “tumor” condition.
Figure 5: Visually there is hardly any difference between the intensities of the xenograft signal in these images.
We thank Reviewer 4 for the suggestion. Unfortunately, as mentioned above, poor resolution of images depends on the software of Journal which compresses pictures when creating the file for peer review. We attach single original pictures with appropriate dimensions and high resolution, so that hopefully the difference is more appreciable.
Round 2
Reviewer 1 Report
Authors have revised this manuscript according to reviewers' suggestions.
Author Response
We thank Reviewer 1 for the endorsement of the revised manuscript.
Reviewer 4 Report
Really nice graphical abstract and the manuscript has improved a lot. I still cannot evaluate the figures as they are way to small (both in general (size) as in resolution.
Additionally, whilst Ct values "per se" do not say anything, really high Ct values (e.g. 38-40 range or higher) indicate a really low abundance of transcripts and unless the comparison value has a really low Ct value (present/absent determination) a subsequent foldchange calculation is pretty much meaningless. Why not mention (in addition to the foldchange) the Ct values?
Author Response
I still cannot evaluate the figures as they are way to small (both in general (size) as in resolution.
We apologize for the inconvenience. We now upload a zipped file containing all figures in their original format, with appropriate size and high resolution.
Additionally, whilst Ct values "per se" do not say anything, really high Ct values (e.g. 38-40 range or higher) indicate a really low abundance of transcripts and unless the comparison value has a really low Ct value (present/absent determination) a subsequent foldchange calculation is pretty much meaningless. Why not mention (in addition to the foldchange) the Ct values?
We agree with the comment that very high Ct values are indicative of absence of transcript, and that in that case FC calculation is useless. We checked that all Ct for each gene and sample analyzed were ≤35 (with only two exceptions in DF1, as reported in the Supplementary Table 1). However, we added Ct values in the Supplementary Table 1 as suggested.